# Caloric Effect Due to the Aharonov–Bohm Flux in an Antidot

**DOI:** 10.3390/nano13192714

**Published:** 2023-10-06

**Authors:** Patricia Martínez-Rojas, M. Esperanza Benavides-Vergara, Francisco J. Peña, Patricio Vargas

**Affiliations:** 1Departamento de Física, CEDENNA, Universidad Técnica Federico Santa María, Av. España 1680, Valparaíso 11520, Chile; patricio.vargas@usm.cl; 2Departamento de Física, Universidad Técnica Federico Santa María, Av. España 1680, Valparaíso 11520, Chile; maria.benavides@sansano.usm.cl (M.E.B.-V.); francisco.penar@usm.cl (F.J.P.); 3Millennium Nucleus in NanoBioPhysics (NNBP), Av. España 1680, Valparaíso 11520, Chile

**Keywords:** magnetocaloric effect, quantum dot, Aharonov–Bohm

## Abstract

In this work, we report the caloric effect for an electronic system of the antidot type, modeled by combining a repulsive and attractive potential (parabolic confinement). In this system, we consider the action of a perpendicular external magnetic field and the possibility of having an Aharonov–Bohm flux (AB-flux) generated by a current passing through a solenoid placed inside the forbidden zone for the electron. The energy levels are obtained analytically, and the model is known as the Bogachek and Landman model. We propose to control the caloric response of the system by varying only the AB-flux, finding that, in the absence of an external magnetic field, the maximization of the effect always occurs at the same AB-flux intensity, independently of the temperature, while fixing the external magnetic field at a non-zero value breaks this symmetry and changes the point where the caloric phenomenon is maximized and is different depending on the temperature to which the process is carried. Our calculations indicate that using an effective electron mass of GaAs heterostructures and a trap intensity of the order of 2.896 meV, the modification of the AB-flux achieves a variation in temperature of the order of 1 K. Our analysis suggests that increasing the parabolic confinement twofold increases the effect threefold, while increasing the antidot size generates the reverse effect, i.e., a strong decrease in the caloric phenomenon under study. Due to the great diversity in technological applications that have antidots in electronics, the possibility of controlling their thermal response simply by varying the intensity of the internal current inside the solenoid (i.e., the intensity of AB-flux) can be a platform of interest for experimental studies.

## 1. Introduction

Innovation in refrigeration systems represents a current research topic due to our planet’s deep climate crisis [1]. The search for materials to replace the compression gases usually used in such techniques is a focus of interest in the industry [2,3]. Within this framework, a topic strongly linked to the study of materials has been developed recently and corresponds to the so-called caloric effects. These effects have a straightforward concept: if a substance has a control parameter that governs a thermodynamic process, the variation of this parameter will generate a change in the entropy of the system, and this change will be proportional to the heat that could be used to cool or heat another technological device according to the second law of thermodynamics. This is why systems with controllable phase transitions are (in most cases) the most studied, since, in general, in this type of transition, the entropy variation of the system is maximized, which results in an increase in heat and consequently the possibility of further heating or cooling an external system more intensively [3,4]. Among these phenomena, the following effects stand out: magnetocaloric (MC) [5,6,7,8,9,10,11,12,13], electrocaloric (ELC) [14,15,16], elastocaloric (EC) [17,18], and barocaloric (BC) [19,20,21,22]. The first of these effects is due to changes in the external magnetic field on the system, the second due to changes in the electric field, the third due to changes in stress, and the last due to changes in pressure.

The most widely studied caloric effect is the magnetocaloric effect (MCE), which corresponds fundamentally to the temperature variation of a magnetic material due to the change in the external magnetic field applied over the system. It has aroused great interest in the scientific community due to its potential applicability, the versatility in its use, and the widespread study of the magnetic properties of different types of materials at present [23,24,25,26,27,28,29,30,31,32,33,34,35,36,37,38,39,40,41,42,43,44,45,46,47,48,49,50,51,52,53]. We highlight the works associated with high-temperature caloric materials [36], antiferromagnetic and ferromagnetic interactions [23,30,34,44,45], heavy lanthanides [46], Fe-Rh alloys [47], and diamagnetic systems [48,49,50,51,52,53]. We would also like to highlight the work associated with the MCE effect on magnetic systems done by V. Franco et al. [54] and D. Serantes et al. [55] in their studies dealing with magnetic nanowire arrays by applying the magnetic field perpendicular to the nanowire’s axis when starting from the magnetic saturation state at the remanence and varying the applied field well below and above the magnetic anisotropy field value of the material.

The MCE has been studied in a quantum dot array where the material temperature variation due to the external magnetic field change has been analyzed for controllable additional effects: geometric confinement, Zeeman, spin–orbit coupling, electric field, and Rashba effect [56,57]. These effects either enhance the thermal effect or cause the system to respond directly or inversely, making it an ideal platform for technological sensors [58]. A less explored material for caloric effects corresponds to antidots, structures with significant potential for high-density data storage due to the possibility of controlling the domain walls of the system. In simple terms, an antidot is a potential hill inaccessible to 2D electrons [59,60,61,62,63,64]. Technological advances allow these systems to work even below T=1 K in temperature [65,66,67,68]. In this conceptual framework, we can highlight the work done with Ni magnetic antidot-type structures by M. Salaheldeen et al. [69], where they studied the role and influence of anisotropy on the MCE with this type of material.

A simple model to characterize an antidot is the one proposed by Bogachek and Landman [70], which constitutes a combination of repulsive potential ($U\left(r\right)\propto r^{-2}$) and attractive potential ($U\left(r\right)\propto r^{2}$) leaving the electron confined in a finite region of space. In addition to these potentials of a purely geometrical nature, this model contemplates the action of an external magnetic field perpendicular to the ring-like structure where the electron is located and considers the possibility of having an Aharonov–Bohm flux (AB-flux) in space, which is generated by a current passing through a solenoid located in the center of the system. The AB-flux is associated with a phenomenon known as the Aharonov–Bohm effect, which is linked to the electrodynamic potentials ϕ (scalar potential) and $\vec{A}$ (vector potential) from electromagnetic theory [71]. For many years, it was thought that in the absence of electric and magnetic fields in a defined region of space, there would be no electromagnetic influences on charged particles. However, in 1959, Yakir Aharonov and David Bohm showed that the vector potential can affect the quantum mechanical behavior of the charged particle even if it is moving in a region where the magnetic field is zero. This can be visualized by considering an infinite solenoid through which an electric current flows and generates a uniform magnetic field inside it and a zero field value outside the solenoid. However, the vector potential is non-zero in the region outside and inside the solenoid. This will affect the electron energy levels, making them dependent on the field flux through the solenoid (AB-flux). This result was impossible to obtain by classical electromagnetic theory, and this is the reason that we consider this phenomenon a pure quantum effect. This effect was tested experimentally by interference by Chambers in 1960 (among other groups at the same time) [72]. Although these potentials are not directly measurable, their impact on physical phenomena has been intensely studied in areas such as electronic transport [73,74,75] and thermodynamics [76,77].

In previous work, we found that AB-flux strongly controls the oscillatory behavior of the MCE, thus acting as a control parameter for the cooling or heating of the MCE [78]. Given this previous analysis, a valid question is whether we can obtain a caloric effect associated only with changes in the AB-flux over the system and under which conditions this effect can be quantified, intensified, and measured for the proposed model. Specifically, even in the absence of an external magnetic field, what would be the thermal response of this system to changes in the current inside the solenoid, which is associated with the control of the AB-flux intensity? In this paper, we answer this question and additionally see the effects on this response regarding the controllable parameters in the model: the size of the antidot, the parabolic trap intensity, and the intensity of the external magnetic field (fixed in this study). The paper is organized as follows: first, there will be an introduction to the energy model presented in Section 2, followed by a discussion of the thermodynamics obtained from the canonical partition function and the physical definitions to quantify the caloric effect shown in Section 3. We finally present the results and discussions for the case without an external field and with an external field present on the material in Section 4.

## 2. Energy Model for the Confined Electron

Let us consider the model described by Bogachek and Landman, which corresponds to the description of a system given by an electron in the presence of a repulsive potential $U_{AD}\left(r\right)$, an AB-flux $\left(\Phi_{AB}\right)$, an external magnetic field B, and finally a parabolic potential $U_{D}\left(r\right)$. The total Hamiltonian, which describes the system, is given by
(1)$$\begin{array}{c} \hat{H}=\frac{1}{2m^{*}}{\left(p+eA\right)}^{2}+U_{AD}\left(r\right)+U_{D}\left(r\right). \end{array}$$

Here, $m^{*}$ is the effective electron mass, A is the total vector potential, and the terms $U_{AD}\left(r\right)$ and $U_{D}\left(r\right)$ are given by
(2)$$\begin{array}{c} U_{AD}\left(r\right)=\frac{\zeta }{r^{2}},\;U_{D}\left(r\right)=\frac{1}{2}m^{*}\omega_{0}^{2}r^{2}, \end{array}$$
where the constant ζ is related to the chemical potential μ and the effective radius of the antidot $r_{0}$ given by the relation $\mu =\frac{\zeta }{r_{0}^{2}}$, and where $\omega_{0}$ is the parabolic trap frequency. The total vector potential involves two terms, $A=A_{1}+A_{2}$, where $A_{1}$ is related to the external magnetic field $B=\nabla \times A_{1}$, and $A_{2}$ describes the additional magnetic flux $\Phi_{AB}$ inside the antidot. For the case of an external perpendicular magnetic field along the *z* direction, $B=\hat{z}B$, we can solve the Schrödinger equation in cylindrical coordinates and use the Coulomb gauge (symmetric gauge) to obtain the energy levels for the confined electron given by
(3)$$\begin{array}{c} E_{nm}^{ad}=\hbar \Omega \left(2n+{\left[{\left(m+\alpha \right)}^{2}+a^{2}\right]}^{1/2}+1\right)+\frac{1}{2}\hbar \omega_{c}\left(m+\alpha \right), \end{array}$$
where $\omega_{c}=\frac{eB}{m^{*}}$ is the cyclotron frequency, $\Omega =\omega_{0}{\left(1+{\left(\frac{\omega_{c}}{2\omega_{0}}\right)}^{2}\right)}^{\frac{1}{2}}$ is the effective frequency of the trap, *n*, *m* are the radial and magnetic quantum numbers, and $a^{2}=\frac{2m^{*}\zeta }{\hbar^{2}}=\frac{2m^{*}\mu }{\hbar^{2}}r_{0}^{2}=k_{F}^{2}r_{0}^{2}$ is a constant proportional to antidot radius $\left(r_{0}\right)$, in which $k_{F}$ is the Fermi wave vector of the electron. The values reported for *a* are located in the region of 0≤a≤10 in the original research [70]. The parameter α is defined in the form $\alpha =\frac{\Phi_{AB}}{\Phi_{0}}$, where $\Phi_{0}=\frac{h}{2e}$ is the magnetic flux quantum. The connection between the α parameter and the AB-flux is given by [78]
(4)$$\begin{array}{c} \alpha =\frac{\Phi_{AB}}{\Phi_{0}}=\frac{AH}{\Phi_{0}}=\frac{\pi r_{s}^{2}H}{\Phi_{0}}, \end{array}$$
where $r_{s}$ corresponds to the radius of the solenoid, H is the value of the magnetic field generated by the current inside the same, and $A=\pi r_{s}^{2}$ is the solenoid section area, whose normal vector is parallel to the magnetic field H. We recall that the field H only exists for $0<r\leq r_{s}$ and is zero outside of the solenoid (i.e., for $r>r_{c})$. Thus, for a given α, the intensity of the magnetic field inside the solenoid has the form of $H=\alpha \Phi_{0}/\pi r_{s}^{2}$. Recent technological advances allow the fabrication of nano-solenoids with a radius of $r_{s}=35$ nm, composed of graphene [79].

We can analyze three asymptotic cases from the energy spectrum given by Equation (Equation 3): (a) the case of a quantum dot, (b) the case of a pure antidot, and (c) the Landau case. The (a) case is obtained when α=0 and a=0, which means that Equation (Equation 3) is reduced to the well-known expression for the Fock–Darwin levels given by
(5)$$\begin{array}{c} E_{nm}^{dot}=\hbar \Omega \left(2n+|m|+1\right)+\frac{1}{2}\hbar \omega_{c}m. \end{array}$$

Since a=0 implies ζ=0, the antidot repulsive potential of Equation (Equation 2) vanishes, and the system then corresponds to a quantum dot. The (b) case is obtained when $\omega_{0}\to 0$, and, from Equation (Equation 3), we have that the energy spectrum takes the form
(6)$$\begin{array}{c} E_{nm}^{antidot}=\hbar \omega_{c}\left[n+\frac{{\left[{\left(m+\alpha \right)}^{2}+a^{2}\right]}^{1/2}+\left(m+\alpha \right)+1}{2}\right]. \end{array}$$

Finally, the (c) case can be obtained by applying to Equation (Equation 6) the case of α=0 (vanishing AB-flux) and a=0 (vanishing antidot), finding the energy spectrum given by
(7)$$E_{nm}^{Landau}=\frac{\hbar \omega_{c}}{2}\left(2n+|m|+m+1\right),$$
that corresponds to the Landau levels in cylindrical coordinates. It is essential to mention that the Landau energy levels are strongly degenerate for all negative values of *m*. Nevertheless, the inclusion of the antidot repulsive potential in the form of Equation (Equation 2) causes the energy levels of Equation (Equation 6) to have an asymptotic degeneracy when m→−∞. The discussion of these three cases shows the completeness of the Bogacheck–Landman problem and the versatility that can be obtained to analyze real parameters with experimental applications in mind.

## 3. Calculation of Partition Function and Thermodynamic Functions

We can calculate the partition $Z_{ad}$ function, using the general solution of Equation (Equation 3) and summing over n(n=0,1,2,…) and m=0,±1,±2,…
(8)$$\begin{array}{c} Z_{ad}=\sum_{n,m}e^{-\beta E_{nm}^{ad}}. \end{array}$$

Unfortunately, the structure of the energy levels of Equation (Equation 3) does not allow a full analytical solution, so we use numerical calculations to obtain the canonical partition function of Equation (Equation 8). We separate the contributions of antidot energy $\left(E_{nm}^{ad}\right)$ in the form
(9)$$\begin{array}{ccc} Z & = & \sum_{n}e^{-2\beta \hbar \Omega \left(n+\frac{1}{2}\right)}\sum_{m}e^{-\beta \hbar \Omega {\left[{\left(m+\alpha \right)}^{2}+a^{2}\right]}^{\frac{1}{2}}-\frac{\beta \hbar \omega_{B}}{2}\left(m+\alpha \right)}\\ & = & \frac{1}{2}\mathrm{csch}\left(\beta \hbar \Omega \right)\sum_{m}e^{-\beta \hbar \Omega {\left[{\left(m+\alpha \right)}^{2}+a^{2}\right]}^{\frac{1}{2}}-\frac{\beta \hbar \omega_{B}}{2}\left(m+\alpha \right)}. \end{array}$$

The values that α can take are not a priori-restricted. However, there is a particularity in the energy spectrum given by Equation (Equation 3) that has repercussions for the partition function and, therefore, in the thermodynamic quantities. If α takes integer values N, the partition function will take the same values as α=0. This is because it is always possible (if α is an integer) to write a new quantum number of the form $\tilde{m}=m+\alpha$ and make the sum of the different energy levels of the partition function with the new quantum number. Concerning the range of temperature in our calculations, we work in the range from 0.5 K to 2 K, allowing us to consider the quantum number m=−70 to m=30, which is sufficient to guarantee good convergence in the thermodynamic calculations that are presented in this work.

### Entropy of the System and Caloric Response

In our thermodynamic analysis, it is important to recall that the electronic entropy is derived from the partition function Z. In the generic form,
(10)$$S_{e}\left(T,B,\alpha \right)=k_{B}T{\left(\frac{\partial \ln Z}{\partial T}\right)}_{B,\alpha }+k_{B}\ln Z.$$

On the other hand, the total entropy for this model can be written as
(11)$$S=S_{e}\left(T,B,\alpha \right)+S_{l}\left(T\right),$$
where $S_{l}\left(T\right)$ is the entropy of the lattice related to the contribution of phonons in the system. Equation (Equation 11) assumes the following approximations: the entropy of phonons relies solely on temperature, thus neglecting the influence of phonon coupling with external magnetic fields. Furthermore, for the comprehensive assessment of entropy, distinctly for electrons and phonons, the discussion within this study excludes the consideration of electron–phonon interactions.

To understand the quantification of the caloric phenomenon, we must consider entropy as a state function that depends on three thermodynamic variables: T,B, and α. This means that we have a function for the entropy of the form S≡S(T,B,α). This allows us to write the total differential expression for the entropy as follows:(12)$$\begin{array}{c} dS={\left(\frac{\partial S}{\partial T}\right)}_{B,\alpha }dT+{\left(\frac{\partial S}{\partial B}\right)}_{T,\alpha }dB+{\left(\frac{\partial S}{\partial \alpha }\right)}_{T,B}d\alpha . \end{array}$$

From this last equation, we can take two paths to quantify the effect: (i) an adiabatic trajectory and (ii) an isothermal trajectory. Analyzing an adiabatic-type process implies that Equation (Equation 12) must be zero. This way, we can clear dT and obtain the temperature variation along this process by integration. This temperature variation is called $\Delta T_{ad}$ and is given by the expression
(13)$$\Delta T_{ad}=-\int_{\alpha_{i}}^{\alpha_{f}}\frac{T}{C_{B,\alpha }}{\left(\frac{\partial S}{\partial \alpha }\right)}_{T,B}d\alpha ,$$
where we use that $C_{B,\alpha }=T{\left(\frac{\partial S}{\partial T}\right)}_{B,\alpha }$ corresponds to specific heat at constant B,α parameters, and, in our formulation, we fix the value of the external magnetic field along the process (which means dB=0).

In the case of quantifying the effect employing an isothermal trajectory, which implies dT=0, we obtain from Equation (Equation 12) an entropy variation at constant temperature (ΔS) given by
(14)$$\Delta S=\int_{\alpha_{1}}^{\alpha_{2}}{\left(\frac{\partial S}{\partial \alpha }\right)}_{T,B}d\alpha .$$

If we look at Equations (Equation 13) and (Equation 14), we can find a relationship between these quantities concerning each other. It is found that $-\Delta S\propto \Delta T_{ad}$. Consequently, it is essential to note that when we have a case in which −ΔS>0, we can consider this type of response the direct type. The system will heat up. Meanwhile, when the response is of the −ΔS<0 type, we call this an inverse-type response, and, consequently, the system will cool down. Therefore, we expect, for a direct response, a $\Delta T_{ad}>0$, and, for the case of an inverse response, we expect a $\Delta T_{ad}<0$ for the final result of the caloric phenomena.

It is also possible to quantify the phenomenon by employing explicit calculation without performing the integration presented in Equations (Equation 13) and (Equation 14), proceeding as follows. To obtain the temperature variation ($\Delta T_{ad}$) directly when performing an adiabatic process, we can use a contour plot by applying the following conditions:(15)$$S_{e}\left(T,B_{0},\alpha \right)=S_{e}\left(T_{0},B_{0},\alpha_{0}\right)=\mathrm{cnt}.,$$
where $\alpha_{0}$ corresponds to the initial value of the AB-flux, $T_{0}$ the initial value of the temperature of the process, and $B_{0}$ the fixed value of the external magnetic field over the sample when the process is carried out. In Equation (Equation 15), we use the fact that, for low temperatures, the condition $S_{e}>>S_{l}$ is satisfied, and therefore we have an approximation for the entropy of the form $S∼S_{e}.$ This contour plot will give a plot of *T* vs. α for different values of constant entropy, thus visualizing explicitly how the temperature varies as the AB-flux changes in an adiabatic process.

For the case of the entropy variation at a constant temperature and a constant external magnetic field, the expression can be given by the difference in the entropy at the initial and final points of the process in the following form:(16)$$-\Delta S_{t}=-\Delta S_{e}=S_{e}\left(T_{0},B_{0},\alpha_{0}\right)-S_{e}\left(T_{0},B_{0},\alpha \right),$$
where the contribution of $S_{l}$ has been canceled out due to its dependence only on temperature, and, consequently, in an isothermal process, it will have no associated variations. Equation (Equation 16) will generate for a given $T_{0}$, $B_{0}$, and $\alpha_{0}$ a function that will depend only on the α variable, thus obtaining a direct plot of $-\Delta S_{e}$ vs. α quantifying the effect accordingly. It is important to mention that the results of the proposed caloric phenomenon presented in subsequent sections take the path discussed for Equations (Equation 15) and (Equation 16).

We must note the differences concerning our previous work in Ref. [78]. In that work, a study of the MCE of this model was performed so that the reported temperature variations could be obtained directly through integration over the specific heat at a constant magnetic field and the derivative of magnetization of the system [78]. We, therefore, emphasize that, in this work, the expression for the $\Delta T_{ad}$ presented in Equation (Equation 13) is a caloric effect purely related to the change in AB-flux. This is why we cannot qualify this study as an MCE type.

## 4. Results and Discussion

### 4.1. Caloric Effect without External Magnetic Field

We will begin the discussion of the results in the absence of an external magnetic field with parameters given by $\omega_{d}=4.4$ THz (related to energy of the order of ∼2.896 meV), a=4, and $m^{*}=0.067m_{e}$ (GaAs) [80,81]. In Figure 1, we can observe in panel (a) the contour plot for an isentropic process of the form $S_{e}\left(T,0,\alpha \right)=$ cnt., where we can directly obtain the relation between the temperature and AB-flux. We note from there that the temperature has a decreasing and then increasing behavior, which means that, depending on the initial value that we take as a reference to start the process, the system can be cooled or heated. To visualize this, on the contour plot, we have drawn a trajectory at a constant temperature given by T=1.3 K (red horizontal line in Figure 1a), and we have marked the process of constant entropy in a green color corresponding to $S_{e}\left(1.3,0,0.22\right)$ (Figure 1a). It can be seen that depending on the α region, the initial temperature of the system can be lower or higher depending on the final value that the control parameter will take. All points on the green line above the red line are points where the temperature will always be higher than the initial temperature. In contrast, points on the green line below will result in final temperatures lower than the initial reference temperature given for the substance. Additionally, we find that the maximum effect for this case occurs for α=0.5. We will see later whether this holds for all ranges of model parameters. The maximum temperature variation observed for this case is about ∼1 K (in absolute value) since the temperature at the minimum point of the contour marked by the green line at α=0.5 is T=0.34 K; thus, ΔT=T−1.3 K = −0.96 K.

The temperature variation results discussed from Figure 1a should be consistent with the results that would be obtained for the $-\Delta S_{t}$. Therefore, there should be regions where this result is positive and in others negative. This can be visualized in Figure 1b, where we observe an oscillatory behavior for the entropy variation at a constant temperature as a function of α, obtaining the minimum for all temperatures shown there, at α=0.5. It is also observed that the temperature increase decreases the effect considerably for the same initial conditions. We can observe from Figure 1b that the case for T=1.3 K gives an $-\Delta S_{t}/k_{B}$ of approximately −0.25; for T=1.7 K, an $-\Delta S_{t}/k_{B}$ of −0.13; and for T=2.1 K, an $-\Delta S_{t}/k_{B}$ of 0.07. This shows how sensitive our system is to the effect of increasing temperatures. Extrapolating these results indicates that the entropy variation will tend towards zero at high temperatures, making the caloric response disappear.

For the variation in the antidot size associated with the value of the parameter *a*, we can appreciate from Figure 2a the variation in the entropy as a function of α for equal parameters of initial temperature, fixed external field, and AB-flux, where it is observed that as the size of the antidot increases, the caloric effect decreases. This is because in the energy spectrum given by Equation (Equation 3), α (contained between 0 and 1 in this study) becomes very small compared to the values of *a* presented. Consequently, small variations in $\alpha^{2}$ will not compensate for the quadratic order in the energy of $a^{2}$, thus making the thermal changes less noticeable when using the AB-flux as a control parameter in the model.

Another adjustable parameter corresponds to the frequency of the parabolic trap, $\omega_{0}$. Figure 2b shows the entropy variation as a function of α for different coupling frequencies, for the same process parameters in terms of temperature, external field, and initial value of α. As $\omega_{0}$ increases, the proposed caloric effect also increases. This is consistent with the *a* parameter, where we discussed that as *a* decreases, the caloric effect increases. Consequently, increasing the value of the trap intensity further confines the electrons, and thus both effects, that of decreasing *a* and increasing $\omega_{0}$, lead to analogous results in this model.

To verify that the point where the caloric effect is maximized for the case in the absence of an external magnetic field is independent of the parameter *a*, we have plotted in Figure 3 the contours for the cases of constant entropy for a=3 (a), a=4 (b), a=5 (c) and a=6 (d). It is observed that, independently of the value of the antidot size, the effect is maximized at the same value given by α=0.5. Of course, the intensity of each is different, the most remarkable being that of a=3 presented in panel (a) of Figure 3. Moreover, it reinforces the results discussed above regarding the decrease in the heating effect as *a* increases, where we see that the effect decreases strongly, finding, for example, for the case of a=6 (panel (d) of the Figure 3), contour plots where almost horizontal curves are appreciated, which would mean that there is no temperature variation with changes in the α parameter and therefore the loss of the caloric phenomenon.

This exciting result about the location of the maximum caloric effect in the absence of an external magnetic field can be understood from the entropy plot as a function of the parameter α presented in Figure 4 for different temperatures between 1.8 K and 0.3 K. The entropy for any value of *T* clearly shows a maximum at α=0.5 for all values of *T*. Consequently, the maximum in the caloric phenomenon will always be located at the same point. Of course, the most abrupt change is given for smaller and smaller temperatures. In contrast, the entropy will not show substantial variations around the α parameter as the temperature increases. It will generate horizontal curves, and no caloric effect associated with the deviation of the AB-flux will be reported. According to all the discussions in this subsection, the best configuration to obtain a high caloric response for this system in the absence of an external magnetic field would be associated with a low temperature (T<2 K), a small antidot size (a<3), and high parabolic confinement.

### 4.2. Caloric Effect in the Presence of External Magnetic Field

In this subsection, we will analyze where a (constant) perpendicular magnetic field is present while thermodynamic processes are performed by varying the parameter α. To do so, we will first analyze the case of constant entropy contours for a=4 of $S_{e}\left(T,1,0.22\right)$. In particular, we will fix the external field at 1 T. This case can be observed in Figure 5a, where we appreciate that the maximum effect is now produced at another point different from 0.5. Symmetry breaking was observed for the caloric phenomenon when no external field was present. Moreover, let us consider Figure 5b. We see that the temperature now affects the maximum point of the effect, and, again, contrary to the case without a field, the maximum for a fixed temperature is not seen for all cases at the same point. Specifically, the magnetic field generates a shift in the thermal response and windows of direct and inverse non-symmetrical heating effects to the variation in the proposed control parameter. In addition, not only the location of the maximum is modified, but also the passage from an inverse to a direct response for the caloric effect. Let us compare Figure 2b with Figure 5b, for a=4. The transfer from an inverse to a direct response type for the case without an external field co-occurs for any temperature at the same α parameter space point (with the same initial condition). In contrast, for the case with a magnetic field, this varies depending on the temperature at which the isothermal process occurs. As the temperature increases, the transfer from a direct to an inverse effect requires a more considerable change in the α parameter when the external field is applied. The different points where the heating effect changes can be seen in Figure 5a, where the cases of $S_{e}\left(T,1,\alpha \right)=S_{e}\left(1.3,1,0.22\right)$ (green curve), $S_{e}\left(T,1,\alpha \right)=S_{e}\left(1.7,1,0.22\right)$ (yellow curve), and $S_{e}\left(T,1,\alpha \right)=S_{e}\left(2.1,1,0.22\right)$ (orange curve) and their respective horizontal curves at a constant temperature at T=1.3 K (red line), T=1.7 K (orange line), and T=2.1 K (purple line) are plotted. For example, we have devised the vertical (light blue curve) at α∼0.45, which is the intersection point of the effect for the T=2.1 K case. All the other transition points occur before this value in line with those presented in Figure 5b.

We have plotted the different contour lines for the case of S(T,1,α)= cnt. in Figure 6 for a=3 (panel (a)), a=4 (panel (b)), a=5 (panel (c)), and a=6 (panel (d)). The symmetry breaking in the caloric effect produced by the external field for different values of the antidot size can be seen. It is also corroborated that the effect decreases as the value of *a* increases, in the same way as in the case without an external magnetic field. Regarding the intensity of the effect, there is no increase or decrease in the caloric phenomenon due to the presence of the external field. Therefore, the net effect of a constant magnetic field perpendicular to the material generates a displacement in the space of the α parameter where the change from a direct response to an inverse type and from an inverse to a direct one occurs. This shift in the maximum in the caloric phenomena can be very beneficial since it is possible to find the configuration that maximizes the effect for small α increments (close to zero α). This would mean that, experimentally, less current would be needed to pass through the solenoid, which is cost-effective and reduces the possible additional interactions that the system may have with its environment.

Finally, we would like to emphasize that our material (antidot) is non-magnetic and corresponds to a free-electron-like system (i.e., no magnetic moment is associated with atomic sites as in the case of Fe, Ni, Co, etc.). Thus, quasi-free electrons can be confined in a semiconductor heterostructure, such as GaAs and Al${}_{x}$Ga${}_{1-x}$As, (x = 0.3) [82]. At room temperature, the energy gap of GaAs is 1.43 eV, while it is 1.79 eV for AlAs [82]. Consequently, electrons in GaAs are confined in a one-dimensional potential well in the z-direction. Therefore, the electrons are trapped in a 2D space where a magnetic field along the z-axis can be applied. These electrons are quasi-free, meaning that they are delocalized and can be confined by high magnetic fields, but there are no magnetic moments associated with atomic sites. We apply an external magnetic field and subject the antidot to a magnetic vector potential generated by a solenoid located perpendicular to the system within the antidot hole. This is the Aharonov–Bohm part. Nonetheless, studying a magnetic system with anisotropy as in Ref. [69] from the perspective of caloric processes is highly desirable since, undoubtedly, magnetic anisotropy, such as magnetocrystalline anisotropy and shape anisotropy, among others, additionally has an influence on the system’s energies and must play a crucial role in thermodynamic processes such as those treated in this work.

## 5. Conclusions

In this work, we have presented the possibility of observing a caloric effect associated with the Aharonov–Bohm flux variation in an antidot-type system in a fixed magnetic field perpendicular to the system. In the absence of this external field, we have found that the effect is maximized for all cases treated in the same value of the Aharonov–Bohm flux independently of the temperature that follows the process. In contrast, the presence of the external field substantially modifies the location where the effect is maximized. This modification can be used to maximize the effect to be as close as possible to α=0, which would mean a lower intensity in the current change that would need to pass through the solenoid to measure the same ΔT variation that the system undergoes. We observe that the effects occur at very low temperatures. When we are in the order of ∼3 K, the thermal response is almost completely lost for the case without an external field and with the external field. We additionally report that the temperature variation associated with the entropy change at small antidot sizes is much more significant than at larger ones. We also studied the effect of the intensity of the parabolic trap on the system, showing that the higher the value of the trap, the more significant the caloric response. Finally, we highlight that in the absence of an external magnetic field, in the presence of changes in the intensity of the parameter associated with the Aharonov–Bohm flux, we found temperature variations (around ∼1 K for $\omega_{0}=4.4$ THz). Aharonov–Bohm flux does not enter the phonon entropy of the system (or couples weakly), compared to how an external magnetic field would couple with the phonons. This is an advantage since the phonon entropy is usually dominant in the caloric phenomena compared to the other entropies of the system. Although it is generally modeled as a function that depends only on *T*, it may depend on the parameter that controls the caloric phenomenon. Therefore, it cannot be compensated for and canceled out in an isothermal path, hindering the pure measurement of the entropy temperature variation of the substance due to changes in its control parameter. Based on the discussion above, this proposal can be interesting from the experimental point of view regarding caloric effects and their possible technological applications.

## Figures and Tables

**Figure 1 nanomaterials-13-02714-f001:**
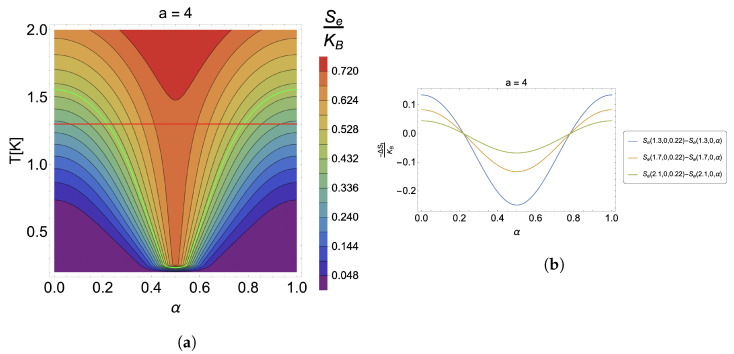
(**a**) Contour plots for the case of S(T,0,α)= cnt. The green line corresponds to the contour of $S_{e}\left(1.3,0,0.22\right)$, and the red horizontal line fixes a temperature of 1.3 K as a reference to quantify the effect. (**b**) $-\Delta S_{e}$ for the case of different temperatures, T=1.3 K (blue line), T=1.7 K (orange line), and T=2.1 K (green line). Here, we have selected the initial value of the α parameter to be 0.22.

**Figure 2 nanomaterials-13-02714-f002:**
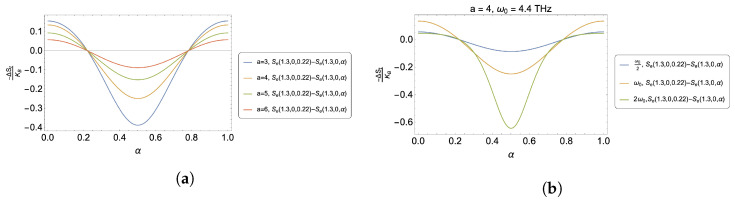
(**a**) $-\Delta S_{t}$ in units of $k_{B}$ for the case of S(1.3,0,0.22)−S(1.3,0,α) for different values of *a*, corresponding to a=3 (blue line), a=4 (yellow line), a=5 (green line), and a=6 (red line). (**b**) $-\Delta S_{t}$.

**Figure 3 nanomaterials-13-02714-f003:**
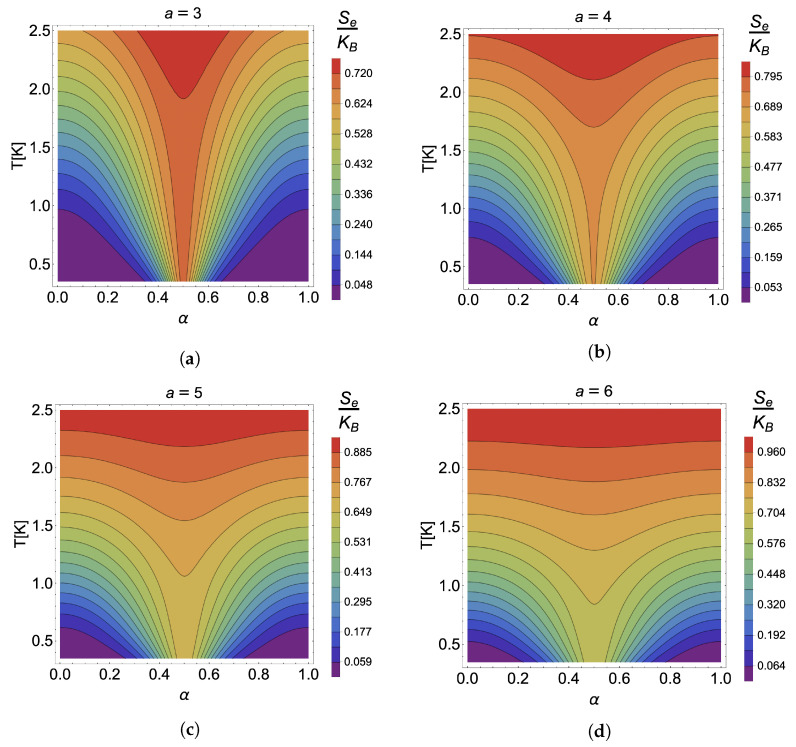
$S_{e}\left(T,0,\alpha \right)=$ cnt. in units of $k_{B}$ for the case of (**a**) a=3, (**b**) a=4, (**c**) a=5, and (**d**) a=6.

**Figure 4 nanomaterials-13-02714-f004:**
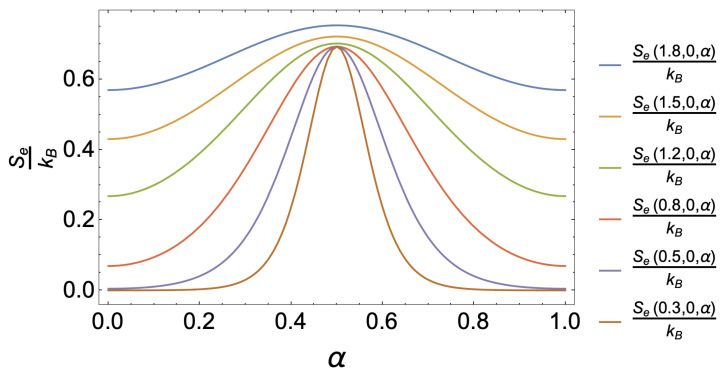
Entropy as a function of the control parameter α for different temperature values between 0.3 K and 1.8 K.

**Figure 5 nanomaterials-13-02714-f005:**
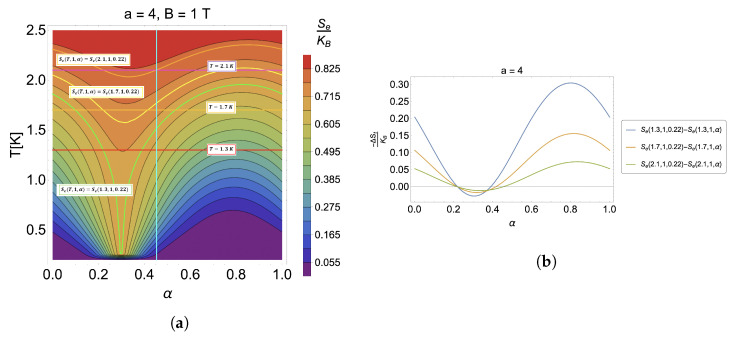
(**a**) Constant entropy contours in the presence of an external magnetic field S(T,1,0.22)= cnt. for the case of a=4. We have marked the cases S(T,1,α)=S(1.3,1,0.22) (green contour line), S(T,1,α)=S(1.7,1,0.22) (yellow contour line), and S(T,1,α)=S(2.1,1,0.22) (orange contour line). We have also plotted the horizontal line of the isothermal process located in T=1.3 K (red horizontal line), T=1.7 K) (orange horizontal line), and T=2.1 K (purple horizontal line). (**b**) Entropy differences for the case of a=4 for T=1.3 K (blue line), T=1.7 K (orange line), and T=2.1 K (green line).

**Figure 6 nanomaterials-13-02714-f006:**
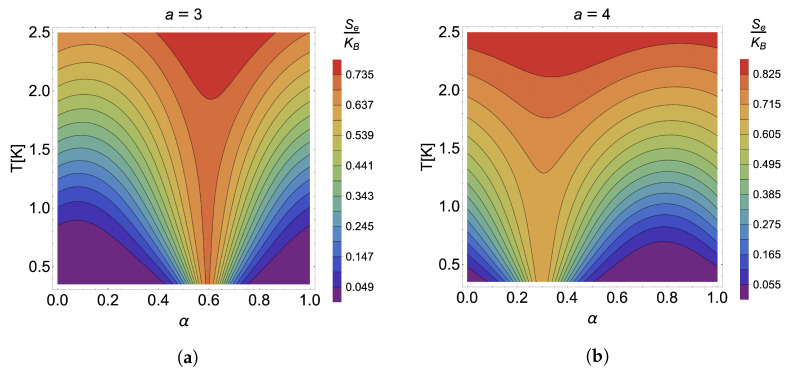
Constant entropy curves in the presence of an external magnetic field S(T,1,α)= cnt. for the case of (**a**) a=3, (**b**) a=4, (**c**) a=5, and (**d**) a=6.

## Data Availability

Not applicable.

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
