# Peer review of "Caloric Effect Due to the Aharonov–Bohm Flux in an Antidot"

_nanomaterials, 2023, doi:10.3390/nano13192714_

Round 1

Reviewer 1 Report

The manuscript titled "Caloric Effect Due to the Aharonov-Bohm Flux in an Antidot" explores the caloric effect in an electronic system modeled as an antidot with a combination of repulsive and attractive potentials. The study investigates the influence of an Aharonov-Bohm flux generated by a solenoid on the system's caloric response in the presence and absence of an external magnetic field.

There are several areas that require improvement and clarification before publication.

1) The abstract provides a brief overview of the research, but it lacks some essential details. It would be beneficial to include specific numerical results or quantitative findings to give readers a clearer understanding of the research outcomes. Additionally, the abstract could be more concise while still conveying the main contributions of the study.

2) The introduction provides a good background on the topic of caloric effects and their importance in refrigeration systems. However, it could be improved by briefly mentioning the significance of the Aharonov-Bohm effect in quantum physics to better introduce its relevance in the context of the study. Moreover, the introduction should clearly state the objectives or research questions addressed in the manuscript.

3) The manuscript describes the energy model for the confined electron in detail, which is essential for readers to understand the calculations. However, the presentation of equations is somewhat dense and may be challenging for readers without a strong background in the field. Consider providing more explanations and context for the equations to enhance readability.

4) Figure 4 lacks detailed labels and captions.

5) The discussion of results should provide more context and interpretation. Explain the physical significance of the observed trends and clarify how the caloric effect changes with variations in parameters such as temperature, antidot size, and magnetic field strength. Discuss the implications of these findings for potential applications or future research directions.

6) The manuscript discusses the caloric effect in the presence and absence of an external magnetic field. It's essential to clarify the definitions of "direct" and "inverse" responses and how they relate to the caloric effect. A more detailed explanation of the observed heating or cooling behavior would be helpful.

7) The conclusion section summarizes the main findings of the study, but it could be expanded to provide a more comprehensive summary. Specifically, the authors should emphasize the practical implications of their research and its relevance to potential applications or further investigations in the field.

The manuscript is generally well-written, but there are some instances where sentence structure or phrasing could be improved for clarity. Ensure that all abbreviations and acronyms are defined upon their first use. Additionally, proofreading for grammar and typographical errors is necessary.

Author Response

Dear Referee, please see the attached document with the answers to your suggestions. These have undoubtedly strengthened the work. 
Best regards,

Francisco J. Peña on behalf of all the authors.

Reviewer 2 Report

Theauthors deal with theoretical approach of magnetocaloric effect (MCE) in antidots system by applying the magnetic field perpendicular to the sample. This peculiar MCE configuration here exposed by the authors is very similar to the studies performed by V. Franco et al, and D. Serantes et al, in their works dealing with magnetic nanowire arrays by applying the magnetic field perpendicular to nanowires axis, when starting from magnetic saturation state, at the remanence, and varying the applied field well below and above the magnetic anisotropy field value of the material, as e. g. in: V. Franco et al,  Physical Review B 77 (10), 104434 (2008); D. Serantes et al, Physical Review B 86 (10), 104431 (2012).

Concerning to the theoretical model, the authors do not explain clearly how it is affected their system by the magnetic anisotropy of the material, if applicable, and in the case of antidots with perpendicular magnetic anisotropy, for example due to its composition, or by varying their geometrical parameters, like in the work recently developed by M. Salaheldeen et al, ACS Applied Nano Materials 2 (4), 1866–1875 (2019), for Ni antidots arrays,  could the authors better explain how can influence the magnetic anysotropy and magnetostatic interaction on the effective MCE response of these antidots system?

English language is well written and clear enough. The text of the manuscript is free of important errors, well organized and can be fluently read by specialized audience.

Author Response

Dear Referee, please see attached document with the answers to your suggestions. These have undoubtedly strengthened the work. 
Best regards,

Francisco J. Peña on behalf of all the authors.

Round 2

Reviewer 2 Report

The authors have carefully revised their manuscript by taking into account the queries of the referees. Their replies to the comments of the referees are well founded and explain the main basis of their teoretical model about the antidots system they study in this work. They have also added more useful information to better clarify the scientific content of their work. Therefore, the manuscript can be considered for its publication in journal in its present form.